# Can We Explain Thousands of Molecularly Identified Mouse Neuronal Types? From Knowing to Understanding

**DOI:** 10.3390/biom14060708

**Published:** 2024-06-15

**Authors:** Luis Puelles, Rudolf Nieuwenhuys

**Affiliations:** 1The Pascual Parrilla Murcia Biomedical Research Institute, University of Murcia, Avda. Buenavista s/n, El Palmar, 30120 Murcia, Spain; 2The Netherlands Institute for Neuroscience, Royal Netherlands Academy of Arts and Sciences, Meibergdreef 47, 1105 BA Amsterdam, The Netherlands; rudolfn@planet.nl

**Keywords:** neuronal cell types, prosomeric brain model, longitudinal zones, neuromeres, progenitor microzonal regionalization, clonal properties, tangential migration, causal explanation

## Abstract

At the end of 2023, the Whole Mouse Brain Atlas was announced, revealing that there are about 5300 molecularly defined neuronal types in the mouse brain. We ask whether brain models exist that contemplate how this is possible. The conventional columnar model, implicitly used by the authors of the Atlas, is incapable of doing so with only 20 brain columns (5 brain vesicles with 4 columns each). We argue that the definition of some 1250 distinct progenitor microzones, each producing at least 4–5 neuronal types over time, may be sufficient. Presently, this is nearly achieved by the prosomeric model amplified by the secondary dorsoventral and anteroposterior microzonation of progenitor areas, plus the clonal variation in cell types produced, on average, by each of them.

## 1. Introduction

The BRAIN Initiative Cell Census Network (BICCN) released the Whole Mouse Brain Atlas publication package in *Nature* on 13 December 2023 (https://www.nature.com/collections/fgihbeccbd, accessed on 5 May 2024). This single-cell transcriptomic, epigenomic, and spatial transcriptomic composite effort updates the number of different neuronal cell types present in the mouse brain to a staggering total of just over 5300, revealing their molecular diversity in concert with their relative positions.

The issues we raise here are: Can we explain how so many different cell types are produced and positioned? This question relates to another: Do we have morphological models allowing the correlation of this level of variety in terms of relative position and neuronal-type specification? The answers are, surprisingly, perhaps and nearly yes.

The BICCN publications implicitly use the conventional columnar brain model of Herrick ([1]; Figure 1a–d), possibly the modified version of Swanson ([2,3]; Figure 1e), or that used by Dong [4] in the Allen Institute’s Adult Mouse Brain Atlas [mouse.brain-map.org]. This model regards the telencephalon, diencephalon, midbrain, hindbrain, and spinal cord as main partitions (five rostrocaudal vesicles; Figure 1a). In this model, Herrick’s smallest units are represented by four functional entities (somatomotor, visceromotor, viscerosensory, and somatosensory columns defined in the brainstem and spinal cord: Sm, Vm, Vs, Ss; Figure 1a,d). This author extrapolated them into the forebrain (i.e., diencephalic Eth, Dth, Vth, Hth; telencephalic Hi, Pir, Str, Se; uniform-colored codes in Figure 1a–c). Note that the forebrain columns possibly perform other functions than the hindbrain ones, though the diencephalon has been interpreted functionally as a continuation of the brainstem. This makes, on the whole, 5 vesicles × 4 columns = 20 columnar units that should produce the recently discovered 5300 neuronal types (average of 265 cell types per column).

The columnar model never postulated any smaller generative parts than the cited columns, and it thus cannot explain today how specific neuronal types are generated by each column (in fact, columns were assumed to have a homogeneous cellular structure throughout; [9]). Relatively recently, the brain was still assumed to contain only a few hundred neuronal types, perhaps close to 1000 [10]. Nevertheless, neuroanatomical studies have identified many discrete neuronal populations within the columns, amply mapped in articles, books, and atlases as nuclear, cortical, or reticular cytoarchitectonic formations. These entities have remained devoid of causal explanation within the columnar model, representing what can be characterized as a ‘potato-sack’ view of an order-less columnar ‘substructure’ (e.g., Krieg’s [11] tridimensional image of the rat hypothalamic column; Figure 2; also consider modern notions about thalamic nuclei, another column). This sort of unexplained hardcore neuroanatomic knowledge (see Figure 2) is so old now (over 100 years) that many think it is normal that we ignore why such a columnar substructure exists.

We know now that a ‘micropotato’ substructure of thousands of neuronal types exists, likewise without an explanation of why each of them emerged and is where it is. They are just mapped by a computer as credible sacks of microscopic neuronal ‘potatoes’ with reproducible boundaries. These ‘micropotatoes’ are more real than the old bigger ones, since they are underpinned by more potent molecular technology, but they seem more difficult to behold than the classic nuclei (we need computers to represent them). It recalls a similar reductive change that occurred when rocks, minerals, and chemicals were reduced to atoms and subatomic particles. That scientific step was no doubt positive in the long run, and the present one may be so likewise for neuroscience. The *neuronal theory* of the brain has been with us now for more than 100 years, pioneered by descriptive efforts by Ramón and Cajal and contemporary colleagues with the Golgi technique ([12,13]; has anyone counted how many different neuron types they described?). Unfortunately, this pioneering theory also lacked causal content and did not convey that there would be thousands of neuronal types. We now see that this matter apparently depends on the developmental combinatorial usage of the genome during neuronal-type specification. We thus should take seriously the issue of causally (reductively) understanding neuronal and glial cell-type multiplicity in the brain, quite apart from the obvious consequences regarding the substantial enrichment of our functional schemata.

During the last century, there has also been a substantial increase in our knowledge of brain connections and functions. Unfortunately, brain connections are likewise largely unexplained. They are, so far, generally unpredictable, except for perhaps those of motoneurons. Helped by experimental hodological studies, lesion studies, and clinical cases, we have learned to schematize some of the basic functional wiring of the brain, a much-valued resource in clinical work (e.g., Figure 3). However, the apparent function of a brain pathway does not tell us how it came to exist ontogenetically or evolutionarily and may obscure less salient parallel functions (see [14]). Conventional hodological schemata refer only to a few cell types, fewer than those really present. Presently, we cannot visualize 5300 brain cell types in our functional diagrams, so much analysis remains to be completed in this context (we need more detailed region-specific diagrams). In the meantime, this leaves us, we posit, with knowledge without complete understanding, not a desirable position in science.

A significant conclusion drawn from the described morpho-functional scenario is that the old-fashioned columnar brain model is irreversibly obsolete. This is due to its excessively simplistic concept of brain generative units (the postulated columns are conceived as homogeneous masses of postmitotic neurons, not even mentioning their progenitors). Moreover, this model remains unable to formulate a molecular developmental causal basis for the nuclear or cortical columnar substructure after 40 years of molecular neurodevelopmental advances, that is, remains devoid of causal developmental hypotheses. If we ever want to understand the why and where of 5300 cell types in the brain, we need much more discriminative molecular and causal evo-devo-friendly brain models.

## 2. Alternative Models

Alternative possibilities already exist. They were recently outlined in “*Towards a New Neuromorphology*” [15] and are likewise flowering in much of the other recent literature. Developmental brain models have been growing in a less simplistic direction, potentiated by a variety of molecular procedures. We can now subdivide the brain more finely than into the classic five vesicles and classify brain progenitors much more discriminatively than in the classical columns. These alternative approaches are already on the verge of being precise and complete enough (in their coverage of all brain territories) to account for thousands of cell types and to specify their corresponding fixed relative positions. The possibility of understanding the brain’s exquisite cellular structure is thus steadily increasing, a scenario that can only benefit functional and pathological analyses. Parallel technical advances in modern molecular hodology, physiology, and neuropharmacology increasingly allow the molecular properties of the neuronal cell membrane and related operative molecules to be translated into biological functions. This is why we answer the first question above with ‘perhaps’ and the second question with ‘nearly so’, given the present partial incompleteness of such clarifying studies.

## 3. Longitudinal Zones of His

The alternative research program that is allowing us to surpass the columnar stalemate deals with mechanisms of developmental brain *regionalization*. This trend possibly started with the work of His in the last quarter of the 19th century [6,7]. This pioneering embryologist discovered longitudinal neurogenetic differences in the embryonic brain wall that led to the differentiation of distinct floor, basal, alar, and roof longitudinal zones (Figure 1f,g and Figure 4a–f). This came jointly with the subsequently corroborated notion of a longitudinal general alar–basal boundary separating alar sensory neuronal classes from basal motor ones. These concepts were inherited by the neuromeric models mentioned next and were validated by accruing molecular results (Figure 1f,g and Figure 4a–f). In the molecular era (the 1980s onwards), we learned that the zones of His result from dorsoventral patterning (DV) triggered by means of sonic hedgehog protein (SHH) diffusing out of the subjacent notochord into the brain primordium. The notochord ends rostrally under the mamillary body (Nch; M; Figure 1f,g). The four longitudinal zones of His run through the whole length of the neural primordium, coinciding at forebrain levels with a *Nkx2.2*-positive alar–basal boundary band (Figure 4b; this approximates the classic limiting sulcus of His, but the band can already be seen at neural plate stages; [16]). These zones end rostrally at the singular acroterminal hypothalamic region, marked selectively by *Dlk1* expression (Figure 4f,g; green-shaded in Figure 1g; this notion was introduced by Puelles et al. [8]). This rostral hypothalamic locus is contacted by the endodermal prechordal plate at the onset of gastrulation and forebrain neural induction, an interaction with a role in AP patterning [17,18]. 

His’s epichordal dorsoventral zones (Figure 4a–g) reflect DV patterning as a causal antecedent of differently specified alar and basal progenitors leading secondarily to Herrick’s [1] brainstem and spinal neuronal columns (Figure 1a); the latter result from secondary microzonal subdivisions of the basal and alar plates: see Figure 5e,f, plus some motoneuronal tangential migrations [20,21]. In contrast, the molecularly validated zones of His contradict Herrick’s and Swanson’s columnar hypotheses for the forebrain [1,2,3] due to their arbitrary forebrain axis ending in the telencephalon, a course objectively not followed by the observable molecularly defined longitudinal zones (Figure 4a–f).

## 4. The Synthetic Neuromeric Models, Leading to the Prosomeric Model

A parallel second step forward resulted from 19th- and 20th-century descriptions of *neuromeres*, that is, series of transverse vesicular bulges of the embryonic neural tube wall (Figure 4b and Figure 5a,b; [5,22,23,24,25,26]). We subsequently learned that neuromeres are specified differentially with partially unique molecular profiles, leading to distinct adult fates. They also share some genetic determinants causing metamery, that is, serial repetition of given features among neighboring units. Some neuromeres clearly substitute previously described ‘columns’ in the forebrain (Figure 1f,g and Figure 4b), and all of them establish an AP subdivision of His’s longitudinal zones (or Herrick’s columns) in the brainstem and spinal cord while retaining the capacity to form modular plurineuromeric neuronal complexes corresponding to classic columns. This pattern allows, in principle, qualitatively different events to happen in each neuromere—e.g., the specification of different cell types—without impeding the functional sensorial or motor unity stressed in the columnar system. The neuromeric pattern thus provides for significant modular cellular and functional diversity and, accordingly, partially reduces the ‘potato-sack’ problem (Figure 1f,g, Figure 4b,c and Figure 5a–c). Moreover, neuromeres can be explained as a result of AP patterning of the entire neural tube (review in [27]). Large tagmata (forebrain, hindbrain, and spinal cord), intermediate proneuromeres (hypothalamus, diencephalon, midbrain, etc.), and final individual neuromeres may be contemplated as molecularly diverse AP constituents (Figure 1g); note that tagmata, proneuromeres, and neuromeres can be still distinguished in the adult (Figure 5a–c).

**Figure 5 biomolecules-14-00708-f005:**
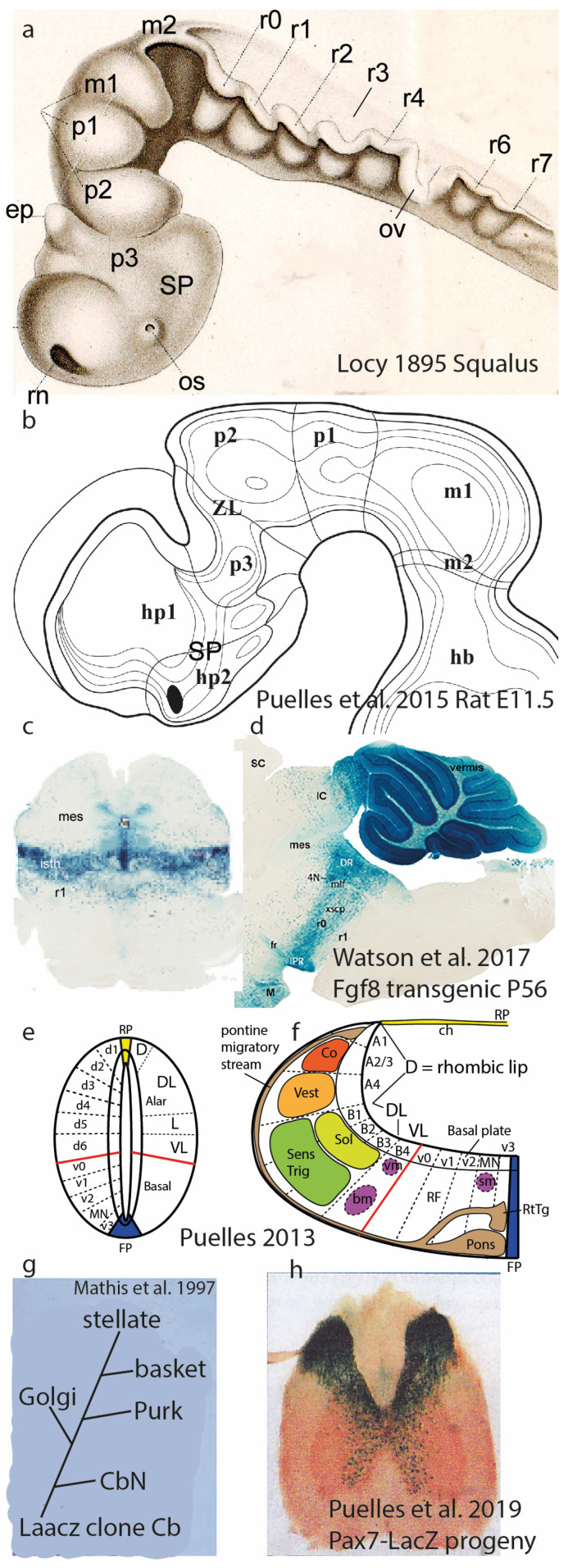
Neuromeres and their DV subdivisions, plus underlying migratory and clonal aspects. (**a**) Forebrain and hindbrain neuromeric bulges in the shark *Squalus acanthias*, drawn by Locy [24]. The forebrain shows prosomeres m1, m2, and p1-p3, plus the secondary prosencephalon (SP). The hindbrain shows 5 preotic rhombomeres comprising prepontine and pontine units (now called r0–r4) and 3 postotic units (r5–r7) corresponding to retropontine (r5,r6) and the first medullary unit (r7). (**b**) Graphic reconstruction of forebrain prosomeres in an E11.5 rat brain, created out of a sagittal section series. The postulated neuromeric cavities of hp1/hp2 (hypothalamus), p1–p3 (diencephalon), and m1/m2 (midbrain) were captured [28]; compare with Figure 4b; see the original for individual sections and correlated descriptions of wholemount AChE-stained postmitotic neurons (also [19]). (**c,d**) Coronal and sagittal sections through transgenic adult mouse brains carrying a Fgf8-LacZ construct that labels all progeny derived from progenitors in which the *Fgf8* gene (a marker typical of the embryonic isthmic rhombomere r0; [29]) was expressed early on. The observed blue LacZ reaction demonstrates that the neuromere-derived adult brain part continues to be transversal, borders the caudal midbrain, participates in the vermal cerebellum, and is complete from the ventricle to the pia; i.e., the neuromere is still there with conserved boundaries but has transformed into its adult counterpart (some blue-labeled cell populations seem to have migrated out of the isthmus into neighboring areas). Analogous material exists for other neuromeres (see [14]). (**e,f**) Schemata of fine microzonal regionalization subdividing the basal and alar domains, observed with molecular markers and experimentation at the spinal cord (**e**) and hindbrain neuromeres (**f**). Usually, there are 5 basal microzones and 6–7 alar ones. Different neuronal populations are produced at each locus. The spinal DL domain (**e**) forms the layered dorsal horn, mixing derivatives from several microzones. In the basal plate, the MN microzone forms the motoneurons, whereas the others form different sorts of interneurons. In the hindbrain (**f**), the alar microzones form the sensory columns (color-coded) and the rhombic lip dorsally, from where most neurons migrate subpially ventralwards, forming, e.g., the pontine and inferior olivary nuclei. Violet-colored motoneurons form three separate columns, two of them migrated into the VL domain of the alar plate (these were wrongly classically classified as a distinct visceromotor column, ignoring that all motoneurons—different subtypes—come from the same MN microzone). (**g**) Radial migration in the cerebellar cortex of a transgenically LacZ-labeled clone composed of different neuron types (cerebellar nuclear: CbN; Purkinje: P; Golgi: G, stellate, basquet), derived from a single, very rare LacZ-reconstitutive event in a progenitor cell prelabeled by a non-functional LaacZ construct [30]. (**h**) Pax7-LacZ progeny derives exclusively from the alar plate, so one expects the spinal cord to contain massive blue derivatives only in the dorsal horn; however, a number of blue alar neurons apparently migrate into the ventral horn (basal plate), probably representing interneurons.

It was long thought that neuromeres are transient early phenomena lacking both discrete derivatives in the adult brain and specific functions, but transgenic technology in the molecular era has shown that their molecular boundaries and derivatives persist even in adults (Figure 5c,d; [29]), though they become otherwise cryptic (invisible by inspection) as the brain wall thickens. Von Kupffer [25], Bergquist and Källén ([5]; Figure 1f), and Vaage [26] listed a number of constant overt (bulging) neuromeres among vertebrates. Later, some non-bulging or cryptic neuromeres showing only a molecular delimitation were added to the ‘prosomeric model’ (Figure 1g). This was the first molecularly based neuromeric model and was studied in the mouse (Figure 1g and Figure 5b; [27,31,32,33,34,35,36,37,38]). Additional neuromere fate-mapping was carried out on the chick, as well as consistent differential gene mappings on chick and mouse embryos, leading to the conclusion that the four dorsoventral zones of His are subdivided into roughly 50 neuromeric segments (7 in the forebrain down to the isthmus, 12–13 in the hindbrain, and over 30 spinal cord units). This already represents a synthesis of 4 His zones × 50 neuromeres = 200 different nominal generative units, theoretically responsible only for an average of 26.5 cell types each. This is surely still too many cell types to be explained as coming from a single generative unit. Neuromeric models divided merely into His primary DV zones are thus also insufficient to solve the present ‘micro-potato-sack’ problem but are no doubt much closer than the columnar models by an order of magnitude (average of 26.5 versus 265 cell types per column).

## 5. Dorsoventral Microzones

Further explanatory possibilities emerged in experimental molecular studies reviewed in [37,39], which indicated that each neuromere does not subdivide dorsoventrally into merely the 4 primary His zones (though these are real; Figure 4), but into 13–14 molecularly singular DV *microzones* per neuromere, each boundary resulting from antagonism between different pairs of ventral and dorsal genes (Figure 5e,f). Each microzone is capable of producing several different neuronal cell types (Figure 5g). We have 1–2 floor plate microzones, 5 basal microzones, 6–7 alar microzones, and 1–2 roof plate microzones. This pattern was first demonstrated in spinal and hindbrain neuromeric units (Figure 5e,f and Figure 6a; [38]) but also appears in the two hypothalamic neuromeres in a particularly expanded manner if the numerous telencephalic subpallial and pallial subdomains are regarded as evaginated alar hypothalamic derivatives (Figure 5b and Figure 6b,c; [8,40,41]). Analogous alar DV microzonal patterning data were reported for the midbrain, pretectum, and prethalamus ([8,42,43]; see Figure 6a). If such a DV microzonal generative substructure were found to apply to all or most neuromeres (unclear yet, but certainly possible), this would allow 13 microzones × 50 neuromeres = 650 distinct generative units instead of the 200 counted above with only 4 His zones. In this case, a theoretical average of 8.15 cell types per DV microzone is reached (we are now already near the needed level of regionalization).

## 6. Anteroposterior Microzones

We will now look at the anteroposterior axis at a much finer scale. At least three forebrain neuromeres have been separately shown to display, in their alar plates, additional AP microzonal molecular regionalization involving an AP tripartition (for cases of the pretectum, see Ferrán et al. [44,45]—Figure 6d–f; for the prethalamus, see Puelles et al. [43]; and for the midbrain m1 mesomere, see [42,46]). Additionally, the thalamus shows at least two AP alar partitions, and a third one is possible ([47]; see legend for Figure 6f). Note that the alar cerebellum in r1 divides into hemispheric, parafloccular, and floccular AP portions. In contrast, the smaller m2 mesomere only shows one AP alar domain [46]. These results accordingly cannot be generalized as yet, pending further studies. In any case, the m2 case may be exceptional due to its small size (Figure 4b). We think that advanced tripartite AP patterning within neuromeric alar fields may yet be found to exist in most neuromeres, allowing more distinct cell types to differentiate (there are also theoretic patterning reasons implying that neuromeres *should have* three AP parts). We thus may tentatively calculate potentially 6 alar DV units × 3 alar AP units × 50 neuromeres = 900 DV/AP alar microzones, which, when added to basal units, reach a new approximate total of 1250 microzones. Note that some brain alar areas, such as the cerebral cortex (in hp1), show disproportionate tangential growth and seem to differentiate into many more than the 3 AP and 6 DV alar microzones of the average neuromere, amounting to some 200 alar cortical areas recently evaluated in the human isocortex [48], without counting other cortical areas, the complex subpallium that produces multiple types of interneurons [40], or the amygdala complex (Garcia-Calero et al. [49]). This potential result of progressive DV and AP microzonal patterning within neuromeric alar domains conceivably extends the theoretical causal explanation to 1250 or more microzones beyond columns and neuromeres. This very rough calculation now implies only the production of 4.4 cell types on average per microzone. We already know several cases of microzones that sequentially produce a few different cell types over time (Figure 5g). 

## 7. Areal Stratification and Clonal Typological Variation

A final important variable in this anti-‘potato-sack’ brain regionalization theory is thus provided by the fourth dimension, *time*. The initial generative potency of microzonal progenitors often changes qualitatively and quantitatively over the local neurohistogenetic period, producing variously stratified or salt-and-pepper-mixed different cell types derived over time from a single microzonal multiclonal progenitor domain (e.g., the diverse retinal cell types, the 6–7 glutamatergic cell types produced sequentially in isocortical areas, or the 6–7 clonal cell types in the cerebellum apart from granule cells; [30]—see Figure 5g). The avian superior colliculus homolog—the optic tectum—forms 14 stratified and morphologically distinct neuronal types. Using the last present calculation, we only need, on average, the generation of 4–5 typologically diversified cell populations in each of the 1250 microzones (some units may have less, while others we know certainly have more) to reach the expected range of 5300 neuronal types.

## 8. Recapitulation and Tangential Migrations

Looking back at our rationale as it applies to the case of the adult hypothalamus, Figure 7 first displays the ‘potato-sack-like’ unclassified and unexplained structure offered by columnar authors (Figure 7a,b; [4,11]). Next, we see AP (neuromeric) and DV (alar and basal microzonal) subdivisions postulated by a neuromeric author (Figure 6b,c and Figure 7c; Puelles’ reference atlas for the P56 Allen Developing Mouse Brain Atlas in 2011; inset c’ is a detail of the alar paraventricular nucleus, showing internal tripartition according to c, consistent with adult *Otp* gene expression). Figure 7d takes the retromamillary basal area within the hp1 neuromere as an example, where we already see two different populations labeled for two gene markers. Their behavior leads to a mixed (fuchsia and green) migration stream of retromamillary neurons, mapped with *Foxa1* (green) and *Nr4a2* (fuchsia), that exit hp1, enter the hp2 neuromere while eschewing invasion of the neighboring mamillary body, reach the ventral tuberal microzone, and, there, form the compact ventral premamillary nucleus (PMV; Figure 7d,e; López-González et al. [50]). The higher-magnification detail in Figure 7e illustrates multiple cell types in this migration (encircled green-, white-, yellow-, fuchsia-, and red-fluorescent cells, indicating various combinations of the two gene markers used, together with the fluorescent migration tag). This multiplicity of PMV cell types was corroborated by single-cell transcriptomic studies discussed in the paper. This highlights surprising partial aspects of the causal cellular origin within a particular hypothalamic classic ‘potato’, the PMV nucleus, which was already known but not explained by Krieg ([11]; Pv in Figure 2). The well-known subthalamic nucleus also results from multitypological ventrodorsal migration, which originates separately from the retromamillary area, though columnar tradition unaccountably does not recognize this nucleus as being hypothalamic (e.g., absent in Figure 7a taken from [11]; see [50]). A single microzone can thus give rise to multiple cell types in different parallel tangential migrations (unless the retromamillary area is actually a pair of microzones; the adjacent mamillary area also displays multiple subnuclei; Figure 7a). The prosomeric model allows further subdivisions.

As occurs in this case, different local neuron types in a microzone may stay aggregated together at a specific stratum or nucleus or may instead mix in various ways with other neighboring cell types within the microzone or a larger domain enclosing primary alar or basal DV areas (when not moving between these; see Figure 5h). This microzone-specific behavior pertaining to the precise positioning of individual neuronal types produced probably involves subtle neuronal adhesive and guidance properties that we still ignore (see the remarkably diverse migrations converging into the prepontine interpeduncular nucleus; [51]). This migrational analysis will probably explain different sorts of cytoarchitectural nuclear or cortical configurations in the future.

A different common phenomenon is that aligned similar neuromeric units with subtle molecular differences may compose a modular motor or sensory column [36,52,53,54,55]. Some cell types may singly or collectively migrate out of their generative microzonal units and neuromeres and functionally incorporate into neighboring microzones, columns, or neuromeres or translocate actively into more distant neural domains. Tangential migrations were once thought to be rare, but we now realize that they occur often, in many parts of the brain, and always in a highly reproducible pattern, meaning they are under molecular control.

The logic for understanding brain-cell-type diversity at many brain positions thus necessarily involves distinguishing local intrinsic neurons from tangentially migrated neurons with more or less distant origins.

## 9. Further Considerations

If the reader is among those desiring the brain to be simple (i.e., less anatomy and more function), our present rationale may ring a bell for caution. If 5300 cell types were to wander around the brain with unconstrained liberty, we would have a truly chaotic ‘potato-sack’ problem for functions, as well. This is not the expected scenario, though, since partial evidence at hand in the Whole Mouse Brain Atlas already indicates that many cell types respect specific boundaries (as in the case of the PMV nucleus; Figure 7d). We only need to understand *molecularly* how these boundaries are first fixed (which *regionalizing mechanisms* operate in the embryo; these are also functions) and then examine how the resulting boundaries are respected (control of *cell migration* and *positioning*; cellular functions). We need more precise molecular identification of operative boundaries and chaos-restrictive guidance effects within and between the, say, 1250 microzones in the brain. This seems a dire perspective, but we have merely 50 neuromeres (most of them in the repetitive spinal cord; only about 20 in the brain). Alternatively, we have just three tagmata: the forebrain, hindbrain, and spinal cord. If the reader wants functional simplicity, generalities noted within a single tagma might be explored. While solid books can probably be written about the functions of any of these large brain domains, even disregarding their intrinsic neuromeric and microzonal phenomena (as anatomists and physiologists have tended to do so far), there might persist a fogginess of conclusions due to a lack of sufficient attention to the complex microzonal participation in those functions. Using a metaphor, if you simplify the inner machinery of your watch too much, the time you read from it will probably be less precise. It is not to your advantage to disregard part of the evolved, tested, and selected structure. If the reader instead prefers to be as close to the truth as possible, the complete microzonal scenario must be explored, as was similarly accomplished with the atomic and subatomic world at the frontier of chemistry and physics (per aspera ad astra). The neuromeres and proneuromeres are just intermediate-level structural brain concepts that possibly explain aspects of axonal guidance, cell migration, and synaptogenesis (see [14] on functionality of neuromeres). Just be wary of columns as sole background concepts or of the thought that, because we have a list of cell types in our computer, we understand the necessary underpinning developmental regionalization or its relationship with function.

The new neuromorphology presented in Nieuwenhuys and Puelles [15] suggests, in essence, replacing unexplained knowledge (e.g., Figure 7a,b) by means of morphological, molecular, and causal pigeonholing of the conventional ‘potato-sack-like’ data sets. This can finally allow us to understand, as a predictable order, the cellular adhesive, connective, and functional complexity that emerges within the central nervous system (Figure 7d,e). This feasible approach should illuminate our perspective on large-scale functional assemblies of neurons and related upper-level psychological phenomena. Eventually, we will glimpse the ethereal ‘butterflies of the soul’ (a poetic Cajalian concept).

## 10. Conclusions

It may thus be postulated that (1) the progressive developmental phenomena of early DV and AP patterning leading to the primary longitudinal zones of His and the crisscrossing series of transverse neuromeres, (2) amplified by the subsequent DV and AP microzonation of neuromeric fields (hierarchy of microzones), (3) plus the temporally patterned clonal neuronal phenotypic variation and differential migration, stratification, and/or aggregation of the cell types produced over time, may well be collectively able to account for the present roughly 5300 distinct neurons of the BICCN consortium and their typical positions.

## Figures and Tables

**Figure 1 biomolecules-14-00708-f001:**
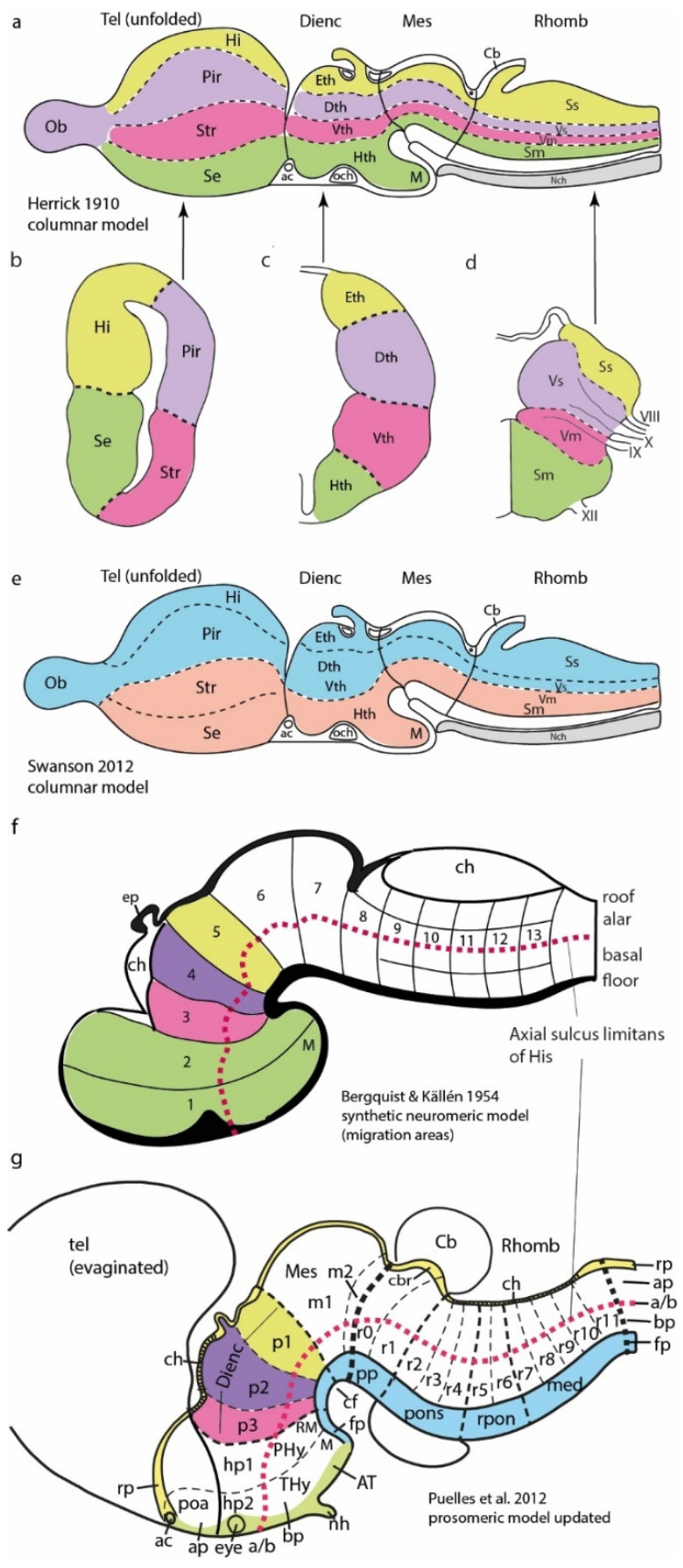
Models mentioned in the text. (**a**) Columnar model of Herrick [1], postulating an unbent length axis ending in the telencephalon. Note that the notochord lies only under the brainstem (Mes, Rhomb), but the postulated longitudinal columns also extend into the forebrain (Dienc, Tel). The rostral ends of the floor and roof domains are not indicated. (**b**–**d**) Cross-sections through Tel, Dienc, and Rhomb in (**a**), illustrating the postulated four columns at each level. (**e**) A modification of Herrick’s columnar model proposed by Swanson [2,3]. Essentially, a general division into basal (cream/pink) and alar (blue) plates is proposed, again without a topographic correlation with the notochord. The roof and floor are not mapped precisely, but the postulated floor possibly reaches the anterior commissure (ac), implying a prechordal part. (**f**) The neuromeric model of Bergquist and Källén [5], also encompassing His’s [6,7] longitudinal zones (names at caudal end) and the alar–basal boundary (red dashes). Note that Tel forms a unit with the hypothalamus, the secondary prosencephalon (green). The diencephalon lies caudal to that and consists of three transverse neuromeres, thought to more faithfully represent the domains with identical color codes in (**a**,**c**). All neuromeres extend from the roof to the floor, whose rostral ends are not represented. The intersection of longitudinal and transversal limits creates a number of quadrangular ‘migration areas’. M indicates the mamillary body. (**g**) The neuromeric (prosomeric) model of Puelles et al. ([8]; significantly updated relative to earlier versions). There are three parallel axial references: (1) the floor (blue) ending rostrally at the mamillary body, (2) the alar–basal boundary (red dashed line; determined molecularly in the forebrain by *Shh* and *Nkx2.2* markers), and (3) the roof plate (yellow), fate-mapped in several vertebrate species to end at the anterior commissure (ac). The concept of a unitary secondary prosencephalon from **f** is maintained, subdivided into two hypothalamo-telencephalic prosomeres (hp1, hp2), whose floor is retromamillary (RM) or mamillary (M). The respective parts of the hypothalamus are named ‘peduncular hypothalamus’ (PHy) and ‘terminal hypothalamus’ (THy). Hp2 ends rostrally at the acroterminal rostromedian domain (green), stretched between the rostral roof and rostral floor, which displays unique formations such as the alar preoptic lamina terminalis and the optic chiasma, and the basal infundibular tuberal region with the neurohypophysis. The diencephalon is trineuromeric, as in (**f**). The midbrain has two unequal neuromeres (m1, m2) and ends at the isthmo-mesencephalic boundary (thick black dashes). The hindbrain shows 12 rhombomeres, some of which are cryptic (detected only molecularly and experimentally). The hindbrain has four proneuromeres (prepontine: pp; pontine: pons; retropontine: rpon; and medulla: med; limits shown by intermediate dashes). The spinal cord (beyond the rhombospinal boundary; thick black dashes) has myelomeres.

**Figure 2 biomolecules-14-00708-f002:**
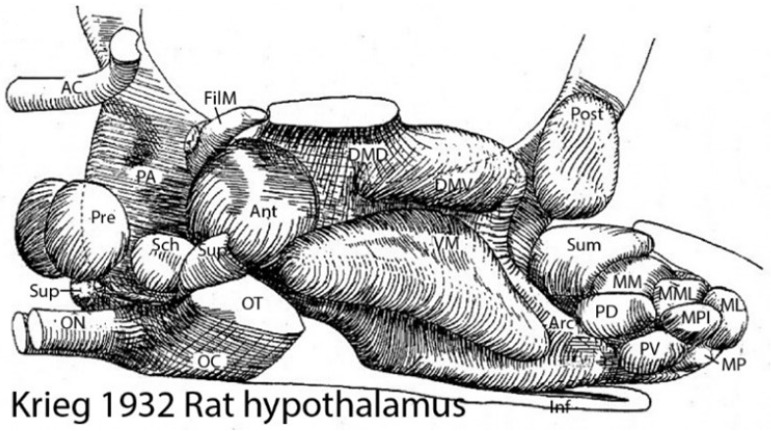
Krieg’s [11] nuclear structure of the rat hypothalamus. Krieg produced this relatively precise 3D view of the hypothalamic nuclear variety in his Figure 21 (note that the hypothalamus was one of Herrick’s diencephalic columns). There is no apparent order in the nuclear arrangement. The columnar caudorostral axis runs from right to left, so the bottom of the reconstruction (Inf) was regarded as the floor of the diencephalon. We see the optic nerve (ON) and the anterior commissure (AC) at the rostral end. AC, anterior commissure; Ant, anterior nucleus; Arc, arcuate nucleus; DMD, dorsal part, dorsomedial nucleus; DMV, ventral part, dorsomedial nucleus; Inf, infundibulum; ML, lateral mamillary nucleus; MM, medial mamillary nucleus; MML, mediomedial mamillary nucleus; MPI, posterior intermediate mamillary nucleus; MP, posterior mamillary nucleus; PA, preoptic area; FilM, filiform (paraventricular) nucleus; PD, dorsal premamillary nucleus; Post, posterior hypothalamic nucleus; Pre, preoptic nucleus; PV, ventral premamillary nucleus; OC, optic chiasma; ON, optic nerve; OT, optic tract; Sch, suprachiasmatic nucleus; Sum, supramamillary nucleus; Sup, supraoptic nucleus; VM ventromedial nucleus.

**Figure 3 biomolecules-14-00708-f003:**
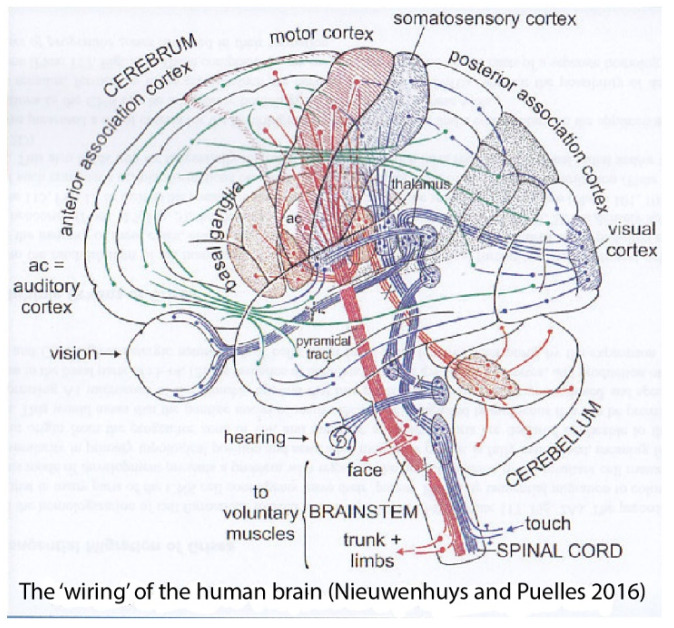
The wiring of the human brain (Figure used in a 1992 talk, reproduced from [15]). A schema displaying sensory and motor pathways jointly with some interconnections.

**Figure 4 biomolecules-14-00708-f004:**
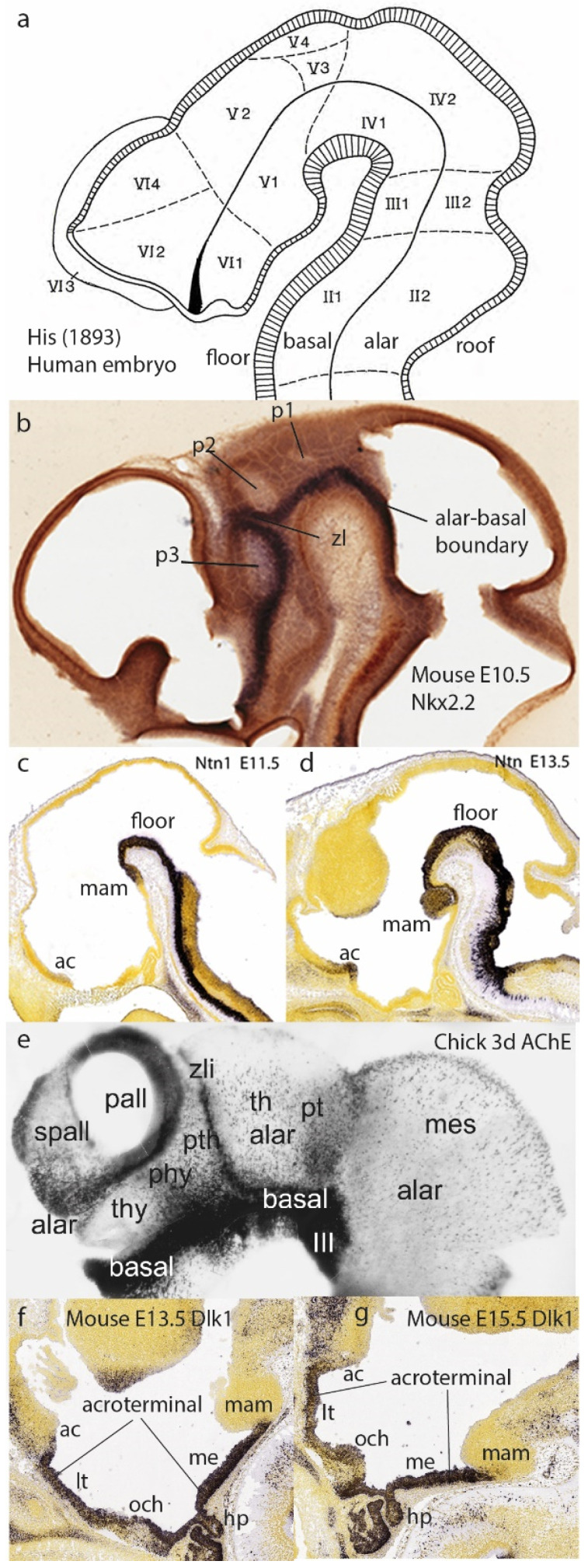
The model of His [6,7] and its DV elements from a modern perspective. (**a**) The schema of His [6,7] with his precise DV subdivision into floor, basal, alar, and roof longitudinal zones (including the limiting and appropriately bent alar–basal boundary sulcus ending at the optic stalk). His also entered relatively imprecise AP partitions (Roman/Arabic tags), now substituted by neuromeres. He was the first to postulate the isthmic domain (tagged III1/III2) as a separate brain segment (see Figure 5c). (**b**) An E10.5 mouse brain showing the alar–basal band of *Nkx2.2* expression (downstream of notochordal SHH signal), which approximates the sulcus of His in (**a**). The neuromeric alar ventricular concavities of the three diencephalic segments are visible (p1, p2, p3). The zona limitans interthalamica (zl) is a transverse singularity of the p2/p3 boundary, caused by a separate *Shh* enhancer, secondarily affecting *Nkx2.2*. (**c**,**d**) Floor plate labeling at E11.5 and E13.5 with *Ntn1* (from [8]; Allen Institute data). The floor stops rostrally over the tip of the notochord at the mamillary pouch (mam). Note also a slight *Ntn1* signal at E13.5 at the locus of the anterior commissure (fate-mapped rostralmost roof plate). (**e**) Three-day-old chick embryo brain reacted wholemount for AChE. This marker identifies postmitotic neurons and is generally negative in progenitor cells (labeling at the zl, the p2/p3 interneuromeric limit, contrarily marks radial glia or progenitors). The neurogenetically precocious basal zone is full of neurons, while the retarded alar zone is relatively unpopulated at this stage. Note some transverse neuromeric borders orthogonal to the alar–basal limit. The telencephalon shows a fully alar pattern, contradicting columnar assumptions ([19]). (**f,g**) The selective expression of *Dlk1* at the mouse acroterminal domain at E13.5 and E15.5 (data from the Allen Developmental Mouse Brain Atlas).

**Figure 6 biomolecules-14-00708-f006:**
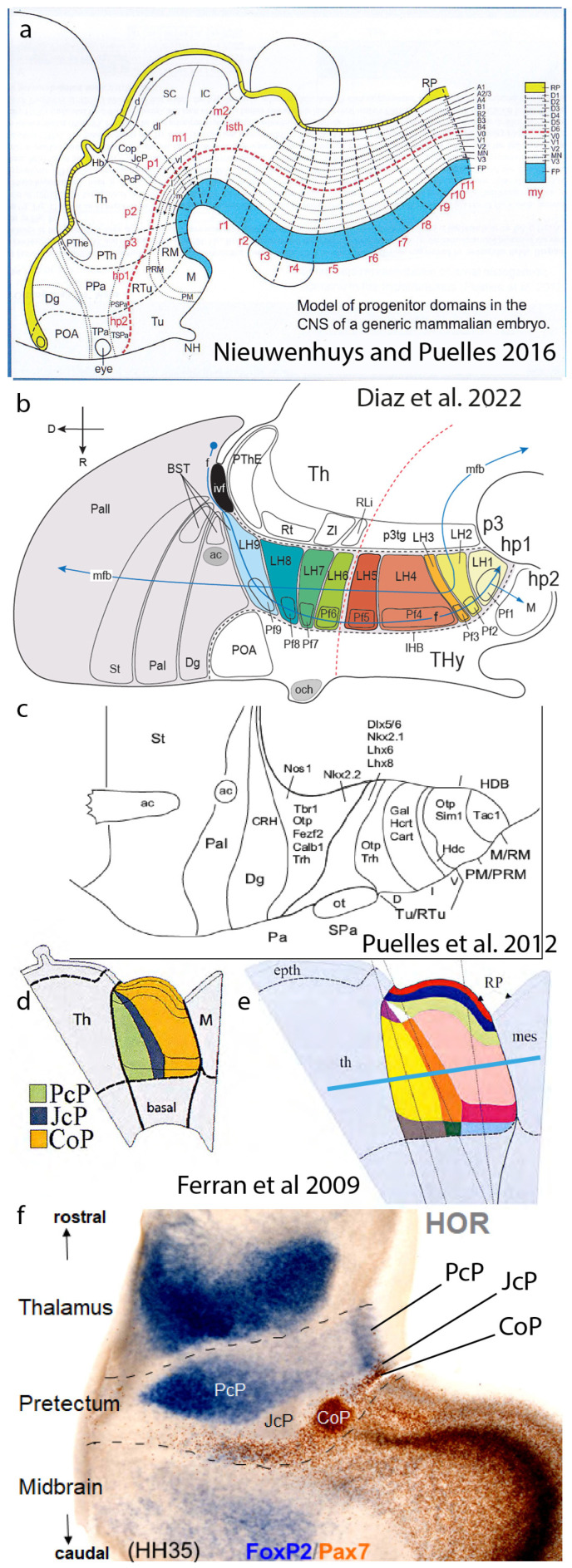
The secondary (advanced) DV and AP microzonal regionalization of neuromeres. (**a**) A prosomeric map of advanced DV patterning reported at different places of the neural tube, collected from the literature and personal results by Nieuwenhuys and Puelles [15]. The mapping was already incomplete at the forebrain levels at that time due to the hypothalamic data of [8] and unrepresented pretectal data from Ferran et al. [44,45]. However, it allows us to appreciate the commonality of the pattern observed in the spinal cord and hindbrain. (**b**,**c**) These panels show partial molecular mapping data in the hypothalamus that suggest a systematic alar and basal DV subdivision into roughly parallel (longitudinal) molecularly diverse microzonal compartments, both at the lateral hypothalamus ((**b**); Diaz et al. [41]) and in the general bineuromeric distribution of peptidergic neurons of several subtypes ((**c**); Puelles et al. [8]). Several single-cell transcriptomic studies corroborate multiple cell types of various sorts in these areas. Our studies correlated this distribution with our molecular characterization of the early progenitor domains. Note that the optic chiasma roughly marks the alar–basal boundary. (**d**–**f**) An illustration of an advanced (secondary) AP microzonal tripartite pattern in the pretectal alar plate (p1 prosomere: precommissural PcP, juxtacommissural JcP, and commissural CoP AP compartments; Ferran et al. [44,45]). The image in (**d**) shows an early stage previous to DV microzonation, while (**e**) displays the added DV subdivision. The image in (**f**) presents a horizontal section from a 9d chick embryo (plane indicated in **e** by a blue bar), where the pretectum (alar p1; boundaries as dashed black lines) is seen lying intercalated between the thalamus (in alar p2) and the midbrain tectum (in alar m1). The *Foxp2* signal selectively characterizes intermediate and superficial strata of the rostral PcP subdomain, while PAX7 immunoreaction likewise marks diverse stratified CoP derivatives; the intermediate JcP subdomain is negative for these two markers but stains specifically, e.g., for *Six3*. This result simultaneously shows the radially complete AP partition and reveals that each microzone forms different neuronal subtypes that occupy different layers of the microzonal mantle. Each of these derivatives is fated to produce different nuclei or cell layers with differential connective properties. The m1 midbrain tectum in this image also shows a more thinly layered rostral microzonal component found next to the CoP and a more thickly layered caudal component (the optic tectum). The signal-free boundary area between the thalamus and PcP may represent the missing third thalamic subdivision (consistent with all neuromeres having this pattern).

**Figure 7 biomolecules-14-00708-f007:**
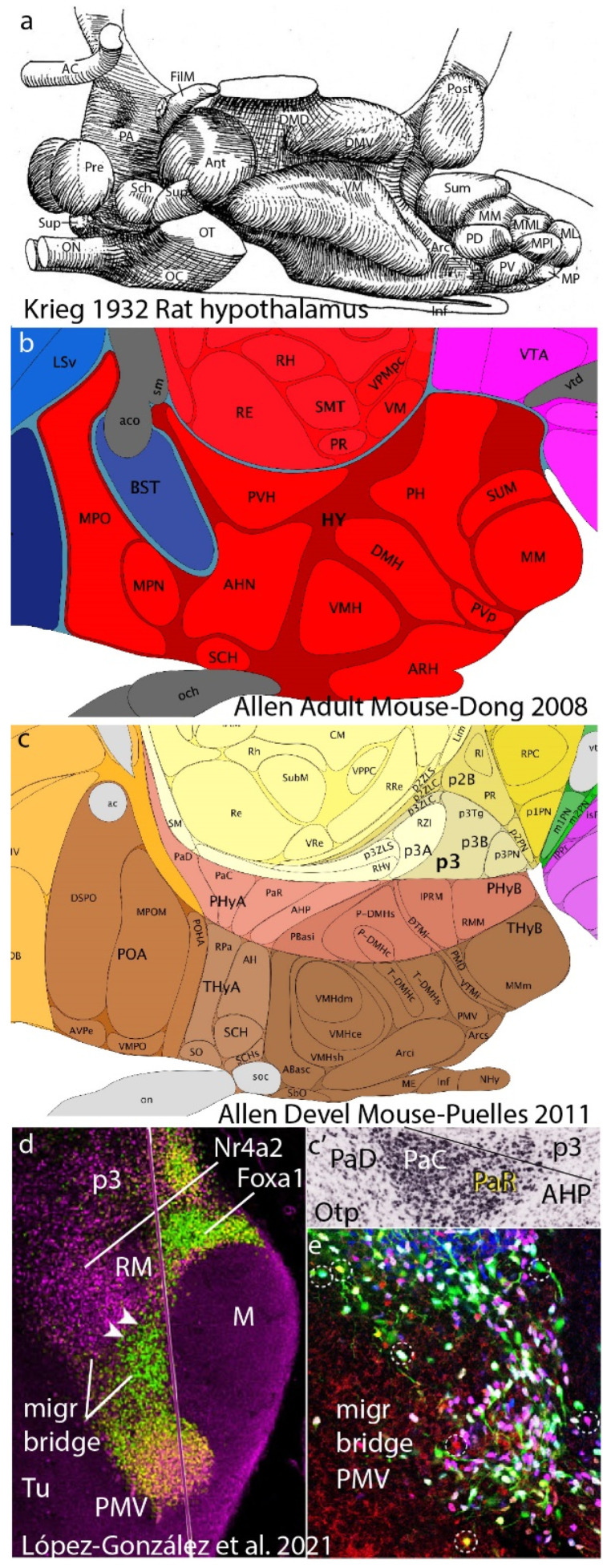
A recapitulation of our argument using the hypothalamus as an example. (**a**,**b**) These panels show two examples of columnar order-less descriptions of nuclei in the hypothalamus, by Krieg [11] in (**a**) and Dong [4] in (**b**), representing examples of what we have called ‘potato-sack morphology’, that is, a description without any sort of classificatory or explanatory concepts. (**c**) Here, LP, working in collaboration with the Allen Institute on the Ontology and Reference Atlases of the Developing Mouse Brain Atlas (2008–2011), proposed the AP partition of the P56 hypothalamus into two hypothalamic neuromeres, defining their respective floor, basal, alar, and roof domains plus DV microzonal divisions. The known adult nuclei fell naturally into the resulting intersectional pigeonholes and were consistent with the gene patterns analyzed. This effort, using some 4000 genes at the Allen Institute, helped to evolve the more complete, updated prosomeric model of Puelles et al. [8]. (**d**,**e**) Jumping now to phenomena in a single microzone from López-González et al. [50], we show a sagittal image of the retromamillary area (RM; a basal plate area of hp1 next to the floor, understood within columnar theory as the ‘supramamillary area’), where we see distinct ventral and dorsal subareas differentially labeled for *Foxa1* (green) and *Nr4a2* (fuchsia). Both populations generate cells that enter a rostralward migration stream that courses above the mamillary body into hp2 and enters the suprajacent ventral tuberal area. As the cells rostrally approach the acroterminal area (which may have attracted them), they stop migrating, forming an oval-shaped nucleus, the classically known ventral premamillary nucleus. There is very little cell mixing with cells derived from hp2, whose cells often express quite different gene markers. In contrast, the PMV migration is composed of intermixed green- and fuchsia-labeled RM cells, which subdivide into several intermixed subgroups according to which of the two markers they express. The final PMV nucleus has a molecularly distinguishable core-and-shell structure with multiple cell types.

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
