# Peer review of "Can We Explain Thousands of Molecularly Identified Mouse Neuronal Types? From Knowing to Understanding"

_biomolecules, 2024, doi:10.3390/biom14060708_

Round 1

Reviewer 1 Report

Comments and Suggestions for Authors

This manuscript performs the heroic service of comparing historical conceptualizations of the organization of the vertebrate brain with the mouse brain BICCN results published at the end of 2023. The authors highlight past errors but also acknowledge early insights. They successfully use our current understanding of nervous system development to show that the reality revealed by the BICCN product does not require the postulation of novel mechanisms but instead demands a profound appreciation of how spatial and temporal developmental mechanisms interact to produce diverse neuronal phenotypes. For those who are not specialists in neuroanatomy, this manuscript provides a relatively concise, well-illustrated, and well-referenced update. And such an update is needed, as we apparently will soon be overtaken by an avalanche of molecularly-based neuroanatomical data (Nature editorial, December 13, 2023). 

I enjoyed reading this manuscript, and I believe that other non-specialists will feel the same. I have minor editorial comments, but only two major style/format comments, which I will describe before listing suggested edits and questions.

First, the second paragraph of the Introduction presents two questions that the authors use to organize the manuscript (and in fact refer back to in a later section). I found these questions vague and hard to understand, and hope the authors will take the opportunity to sharpen them in their revised manuscript. The first question asks how different cell types in the brain are produced and positioned. I think that we already have an understanding of the mechanisms of cell differentiation. Don't the authors really want to ask how so many different cell types are produced during the development of a single organ? The inclusion of the question about positioning at this point in the manuscript - which I see as a secondary issue - seems premature, and I think could be eliminated. The second question asks if we have morphological models capable of meeting the moment, but I was left wondering if the authors were really interested in the spatial nature of development, which made the second question feel redundant to the first. I suggest re-phrasing these questions to focus on their essence and to reduce redundancy (or perceived redundancy).

Second, it is hard to believe that the reader described at the start of the Further considerations section on page 14 (lines 449-450) still exists, given the unequivocal success of modern molecular biology. The only such person I could conjure might be the poor beginning graduate student tasked with mastering a historically-based overview of vertebrate neuroanatomy. Perhaps the authors could address the need to move beyond the teaching of outdated concepts of brain development, if in fact that is still happening. Without any such further development, though, I would recommend beginning this section with the statement that many cell types respect specific boundaries.

Minor comments

Line 25: It would be more accurate to state "a staggering total just over 5300." (to acknowledge the published 5322 figure)

Line 40: I was unclear on the intended meaning of the phrase "harbor other functions." Do the authors mean functions or do they mean subcompartments?

Line 46: substitute long for length

Line 50: the compartment in the figure described as cream is pink  in the review version of the manuscript

Line 60: is the Puelles et al. reference 2012a or 2012b (this is a general problem in the manuscript, and all should be checked and correctly assigned)

Line 64: "the concept of a unitary secondary prosencephalon"

Line 70: replace like with as

Line 79: the Kuhlenbeck 1973 reference is missing from the citation list

Line 84: I like very much the potato sack description, but is it new to these authors? Or is a citation needed?

Line 87: is the phrase in quotation marks a true quotation that needs to be attributed?

Line 115: suggest adding the phrase "on the basis of cytoarchitecture alone" to the question about the number of different neuron types revealed by the Golgi technique

Line 115: premature has slightly negative overtones, implying unformed. Perhaps prescient would be a better word choice?

Line 129: I agree that we cannot visualize 5300 brain cell types on global functional diagrams using tools such as color, but global is perhaps asking too much. Region-specific diagrams would still work, I think.

Lines 136-137: the phrase "drawn morpho-physiological scenario" is hard to understand, but I think I agree completely with this statement. Re-phrase for clarity.

Line 143: perhaps make the point more simply by referring to causal developmental models?

Line 207: there are extra dashes here that are confusing (also Line 225)

Line 220: suggest replacing the phrase badly described with previously described

Line 337: "different direction" seems inappropriate as the AP axis has already been considered at a larger scale. "We will now look at the anterioposterior axis at a much finer scale."

Lines 478-479: I appreciate the beautiful brain of Cajal and think that most neuroscientists feel the same, but the metaphor does not work for me. I've already had my glimpse of the butterflies, now I want to know how the entire ecosystem they live in works. 

Reference 2 (Amat et al., 2022) not cited in the text.

Reference 4 (Bulfone et al., 1993) not cited in the text. 

Reference 20 (Lorente-Canovas et al., 2021) not cited in the text.

References 25 and 26 are out of order. 

Reference 30 (Puelles 2017): I'm unsure this is cited in the text.

Reference 50 (Watson et al 2017a) not cited in the text. 

Kuhlenbeck 1973 and Marin and Puelles 1994 need to be added to the list of citations.

One final general comment: I was surprised that there was no mention of how investigators are tackling the same questions in the Drosophila brain.

Author Response

Reviewer 1

Open Review

(x) I would not like to sign my review report

( ) I would like to sign my review report

Quality of English Language

( ) I am not qualified to assess the quality of English in this paper

( ) English very difficult to understand/incomprehensible

( ) Extensive editing of English language required

( ) Moderate editing of English language required

( ) Minor editing of English language required

(x) English language fine. No issues detected

Is the work a significant contribution to the field?             

Is the work well organized and comprehensively described?       

Is the work scientifically sound and not misleading?        

Are there appropriate and adequate references to related and previous work?   

Is the English used correct and readable?             

Comments and Suggestions for Authors

This manuscript performs the heroic service of comparing historical conceptualizations of the organization of the vertebrate brain with the mouse brain BICCN results published at the end of 2023. The authors highlight past errors but also acknowledge early insights. They successfully use our current understanding of nervous system development to show that the reality revealed by the BICCN product does not require the postulation of novel mechanisms but instead demands a profound appreciation of how spatial and temporal developmental mechanisms interact to produce diverse neuronal phenotypes. For those who are not specialists in neuroanatomy, this manuscript provides a relatively concise, well-illustrated, and well-referenced update. And such an update is needed, as we apparently will soon be overtaken by an avalanche of molecularly-based neuroanatomical data (Nature editorial, December 13, 2023).

I enjoyed reading this manuscript, and I believe that other non-specialists will feel the same. I have minor editorial comments, but only two major style/format comments, which I will describe before listing suggested edits and questions.

First, the second paragraph of the Introduction presents two questions that the authors use to organize the manuscript (and in fact refer back to in a later section). I found these questions vague and hard to understand, and hope the authors will take the opportunity to sharpen them in their revised manuscript. The first question asks how different cell types in the brain are produced and positioned. I think that we already have an understanding of the mechanisms of cell differentiation. Don't the authors really want to ask how so many different cell types are produced during the development of a single organ?

Answer: we entered ‘so many’ as suggested.

 The inclusion of the question about positioning at this point in the manuscript - which I see as a secondary issue - seems premature, and I think could be eliminated. The second question asks if we have morphological models capable of meeting the moment, but I was left wondering if the authors were really interested in the spatial nature of development, which made the second question feel redundant to the first. I suggest re-phrasing these questions to focus on their essence and to reduce redundancy (or perceived redundancy).

Answer: We already say that the second question relates to the first. It emphasizes that different cell types also have specific positions and boundaries, an aspect that likewise needs explanation. This is no redundancy. For resolving that, brain models are needed that postulate and justify how many different parts (=positions) exist in the brain. We changed the word ‘visualization’ into ‘correlation’.

Second, it is hard to believe that the reader described at the start of the Further considerations section on page 14 (lines 449-450) still exists, given the unequivocal success of modern molecular biology. The only such person I could conjure might be the poor beginning graduate student tasked with mastering a historically-based overview of vertebrate neuroanatomy. Perhaps the authors could address the need to move beyond the teaching of outdated concepts of brain development, if in fact that is still happening. Without any such further development, though, I would recommend beginning this section with the statement that many cell types respect specific boundaries.

Answer: Both authors have met colleagues that protested because our detailed embryological explanations were ‘too complex’, meaning too many anatomic details were given and had to be learnt; they did not say they thought we were wrong, but asked for ‘simpler’ schemata, related to more straightforward functional aspects (“less anatomy and more function”). This sort of naïve mentality does exist, since the era of molecular biology has not in general incited neurobiologists to study and advance anatomy. Neuroscience treatises in use have a rather poor, uncritical coverage of that field, and many feel that any detail exceeding that level is ‘excessive’ and can be disregarded. The structure of the brain seems established on what was known 100 years ago, though novel data of interest have continued to accrue. We refer to this scenario because it is relevant in the present context. How often does a normal neuroscientist use the concepts of neuromeres or microzones in his/her work? The answer is rather insignificant. Can we reduce 5,300 different cells types to a simple morphological causal hypothesis that does not involve such normally discarded concepts? We argue that probably not. This is a message that needs to be passed on. In this place we have modified several sentences, including a comparison with a watch whose inner structure you arbitrarily ‘simplify’, trying to make our meaning clearer.

Minor comments

Line 25: It would be more accurate to state "a staggering total just over 5300." (to acknowledge the published 5322 figure)

Answer: Done

Line 40: I was unclear on the intended meaning of the phrase "harbor other functions." Do the authors mean functions or do they mean subcompartments?

Answer: We changed ‘harbor’ to ‘perform other functions than the hindbrain ones’. Also changed in the next line ‘studied’ into ‘interpreted’.

Line 46: substitute long for length

Answer: We think that ‘length axis’ is precisely the classic concept we want to use here, implying an unique, all-important landmark, established by Nature in a particular way, not some less important and variable (arbitrary) reference that merely happens to be ‘long’. It is often wrongly thought in neuroanatomy that the axis is subjective and does not need to be expressly defined. We do not agree with that viewpoint, since the brain length axis is genomically controlled. See the correlation of our roof, alar-basal boundary, and floor longitudinal landmarks with the notochord length in Fig.1a,e, referred to Fig.1g.

Line 50: the compartment in the figure described as cream is pink  in the review version of the manuscript

Answer: We added the pink character as ‘cream/pink’.

Line 60: is the Puelles et al. reference 2012a or 2012b (this is a general problem in the manuscript, and all should be checked and correctly assigned)

Answer: Done

Line 64: "the concept of a unitary secondary prosencephalon"

Answer: Done

Line 70: replace like with as

Answer: Done

Line 79: the Kuhlenbeck 1973 reference is missing from the citation list

Answer: Corrected

Line 84: I like very much the potato sack description, but is it new to these authors? Or is a citation needed?

Answer: This is a personal expression that LP has used in conversation with RN for many years, but this is the first time it has been inserted into a publication.

Line 87: is the phrase in quotation marks a true quotation that needs to be attributed?

Answer: We guess that the reviewer refers to the phrase in quotation marks in line 89. This is no particular quotation (just our description). We eliminated the quotation marks.

Line 115: suggest adding the phrase "on the basis of cytoarchitecture alone" to the question about the number of different neuron types revealed by the Golgi technique

Answer: Cytoarchitecture is technically related to the staining of Nissl granules (polyribosomes) in the neuronal cytoplasm, whereas the Golgi technique generates an indiscriminate aleatory filling of the cytoplasm of specific neurons (including their dendritic and axonal processes) with silver dichromate microcrystals (a dark-brown precipitate). They are thus non-comparable as procedures. The crucial work done by Cajal belongs to the field of Golgi studies and not to cytoarchitectonic analysis. Golgi impregnations show neuronal types in the general shape of the arborizations visualized in complete or partial detail, as well as in the size, form and positions of the cell bodies. We accordingly did not accept this change. We did change in line 117 ‘neurons’ into ‘different neuron types’.

Line 115: ‘premature’ has slightly negative overtones, implying unformed. Perhaps prescient would be a better word choice?

Answer: We changed ‘premature’ into ‘pioneering’.

Line 129: I agree that we cannot visualize 5300 brain cell types on global functional diagrams using tools such as color, but global is perhaps asking too much. Region-specific diagrams would still work, I think.

Answer: We changed accordingly several sentences: “Conventional hodological schemata refer only to a few cell types, less than those really present. At this moment, we cannot visualize 5,300 brain cell types in our functional diagrams, so that much analysis remains to be done in this context (we need more detailed region-specific diagrams).”

Lines 136-137: the phrase "drawn morpho-physiological scenario" is hard to understand, but I think I agree completely with this statement. Re-phrase for clarity.

Answer: Changed the phrase as follows: “A significant conclusion drawn from the described morpho-physiological scenario is ...”.

Line 143: perhaps make the point more simply by referring to causal developmental models?

Answer: Changed phrase as: “Moreover, this model remains unable to formulate a molecular developmental causal basis for nuclear or cortical columnar substructure after 40 years of molecular neurodevelopmental advances, that is, remains devoid of causal developmental hypotheses.

Line 207: there are extra dashes here that are confusing (also Line 225)

Answer: We believe the reviewer refers actually to present lines 202-203, which is the only place with dashes in this legend. For simplicity, we changed the parenthesis to:  “(labeling at the zl, the p2/p3 interneuromeric limit, contrarily marks radial glia or progenitors)”

Line 220: suggest replacing the phrase ‘badly described’ with ‘previously described’

Answer: Done

Line 337: "different direction" seems inappropriate as the AP axis has already been considered at a larger scale. "We will now look at the anterioposterior axis at a much finer scale."

Answer: Done

Lines 478-479: I appreciate the beautiful brain of Cajal and think that most neuroscientists feel the same, but the metaphor does not work for me. I've already had my glimpse of the butterflies, now I want to know how the entire ecosystem they live in works.

Answer: This is a famous Cajal expression that is frequently cited in the sense intended here.

Reference 2 (Amat et al., 2022) not cited in the text.

Answer: Added in line 206.

Reference 4 (Bulfone et al., 1993) not cited in the text.

Answer: Added in line 279.

Reference 20 (Lorente-Canovas et al., 2012) not cited in the text.

Answer: Added in lines 415-417 as: “(see remarkable diverse migrations converging into the prepontine interpeduncular nucleus; Lorente-Cánovas et al. 2012)”

References 25 and 26 are out of order.

Answer: We reordered  all references according to the journal rules (numbering them in brackets according to their order of appearance in the text).

Reference 30 (Puelles 2017): I'm unsure this is cited in the text.

Answer: Added in line 281.

Reference 50 (Watson et al 2017) not cited in the text.

Answer: Added in line 274.

Marin and Puelles 1994 needs to be added to the list of citations.

Answer: Done

One final general comment: I was surprised that there was no mention of how investigators are tackling the same questions in the Drosophila brain.

Answer: Given that our text comments specifically on recent results published on the mouse brain, it did not seem relevant to connect with the Drosophila literature (on which we are not expert, by any means). In any case, the general understanding of developmental patterning and regionalizing processes that occur in insects suggests that the basic genomic mechanisms are very similar -e.g., see “The Origin of Order” by S.A.Kauffman (1993; Oxford Univ.Press). We are not aware, though, of a comparable diversity of causal models of the insect brain, nor of a counting of the number of different neuronal types in those animals.

Reviewer 2 Report

Comments and Suggestions for Authors This work outlines a significant advancement in our understanding of the complexity of the brain, as revealed by the Whole Mouse Brain Atlas. The identification of approximately 5,300 molecularly defined neuronal types of challenges conventional brain models to account for such diversity. The authors propose a model that fits best with the recent discovery that there are 5300 molecular defined neurons. There are several neuroanatomic models that try to explain the structure and organization of the brain. Here, the authors perform a revision of different models describing its limitations to explain the big neuronal diversity. The authors present a compelling argument for the adoption of this new neuromorphology model that incorporates molecular and developmental studies to better understand the complex organization of the brain. The proposal of the microzonal areas of the brain if a new relevant idea to the field because, as the authors write, we should not see the brain as a “chaotic potato-sack” but understand the boundaries and microzonal organization of the brain to better understand its structure and function.

In general, the manuscript is well written, containing the relevant references and the figures help a lot to visualize the models and ideas explained through the text.

Author Response

Reviewer 2

Open Review

Quality of English Language

( ) I am not qualified to assess the quality of English in this paper
( ) English very difficult to understand/incomprehensible
( ) Extensive editing of English language required
( ) Moderate editing of English language required
( ) Minor editing of English language required
(x) English language fine. No issues detected

Is the work a significant contribution to the field?

Is the work well organized and comprehensively described?

Is the work scientifically sound and not misleading?

Are there appropriate and adequate references to related and previous work?

Is the English used correct and readable?

Comments and Suggestions for Authors

This work outlines a significant advancement in our understanding of the complexity of the brain, as revealed by the Whole Mouse Brain Atlas. The identification of approximately 5,300 molecularly defined neuronal types of challenges conventional brain models to account for such diversity. The authors propose a model that fits best with the recent discovery that there are 5300 molecular defined neurons. There are several neuroanatomic models that try to explain the structure and organization of the brain. Here, the authors perform a revision of different models describing its limitations to explain the big neuronal diversity. The authors present a compelling argument for the adoption of this new neuromorphology model that incorporates molecular and developmental studies to better understand the complex organization of the brain. The proposal of the microzonal areas of the brain if a new relevant idea to the field because, as the authors write, we should not see the brain as a “chaotic potato-sack” but understand the boundaries and microzonal organization of the brain to better understand its structure and function.

In general, the manuscript is well written, containing the relevant references and the figures help a lot to visualize the models and ideas explained through the text.

Submission Date

07 May 2024

Date of this review

28 May 2024 12:22:33

Answer: We appreciate these comments.

Reviewer 3 Report

Comments and Suggestions for Authors

This review article by Luis Puelles and Rudolf Nieuwenhuys investigates which brain model that would best accomodate the recently proposed 5300 molecularly defined neuronal types in the mouse brain. The conclusion is that the columnar model, which was used by the Brain Network represents a suboptimal reference foundation. By meticulous exploration of other models, the review authors conclude that a modified version (including spatial (D-V; A-P) and temporal patterning) of the prosomeric model is a better fit. The only comment I have is to better highlight that the Brain Atlas and the review concerns the brain, although the spinal cord is mentioned in each of the brain models, and it would be appropriate to speculate on the number of total neurons in the entire central nervous system would all parts be included.

Author Response

Reviewer 3

Open Review

Quality of English Language

( ) I am not qualified to assess the quality of English in this paper
( ) English very difficult to understand/incomprehensible
( ) Extensive editing of English language required
( ) Moderate editing of English language required
( ) Minor editing of English language required
(x) English language fine. No issues detected

Is the work a significant contribution to the field?

Is the work well organized and comprehensively described?

Is the work scientifically sound and not misleading?

Are there appropriate and adequate references to related and previous work?

Is the English used correct and readable?

Comments and Suggestions for Authors

This review article by Luis Puelles and Rudolf Nieuwenhuys investigates which brain model that would best accomodate the recently proposed 5300 molecularly defined neuronal types in the mouse brain. The conclusion is that the columnar model, which was used by the Brain Network represents a suboptimal reference foundation. By meticulous exploration of other models, the review authors conclude that a modified version (including spatial (D-V; A-P) and temporal patterning) of the prosomeric model is a better fit. The only comment I have is to better highlight that the Brain Atlas and the review concerns the brain, although the spinal cord is mentioned in each of the brain models, and it would be appropriate to speculate on the number of total neurons in the entire central nervous system would all parts be included.

Answer: We are not sure that the Brain Atlas excludes the spinal cord. In any case, our quantitative arguments as regards brain models systematically took into consideration the whole brain including the spinal cord. The only variable is that the number of spinal neuromeres changes with the species (as a result of variable clonal lengthening), something which does not occur with the brain neuromeres.

Submission Date

07 May 2024

Date of this review

02 Jun 2024 15:45:15